# Recent Advances in Hepatocellular Carcinoma Treatment with Radionuclides

**DOI:** 10.3390/ph15111339

**Published:** 2022-10-28

**Authors:** Ruiqi Liu, Hong Li, Yihua Qiu, Hongguang Liu, Zhen Cheng

**Affiliations:** 1Institute of Molecular Medicine, College of Life and Health Sciences, Northeastern University, Shenyang 110000, China; 2State Key Laboratory of Drug Research, Molecular Imaging Center, Shanghai Institute of Materia Medica, Chinese Academy of Sciences, Shanghai 201203, China; 3Shandong Laboratory of Yantai Drug Discovery, Bohai Rim Advanced Research Institute for Drug Discovery, Yantai 264117, China

**Keywords:** hepatocellular carcinoma, transarterial radioembolization, radioactive seed implantation, radioimmunotherapy

## Abstract

As the third leading cause of cancer death worldwide, hepatocellular carcinoma (HCC) is characterized by late detection, difficult diagnosis and treatment, rapid progression, and poor prognosis. Current treatments for liver cancer include surgical resection, radiofrequency ablation, liver transplantation, chemotherapy, external radiation therapy, and internal radionuclide therapy. Radionuclide therapy is the use of high-energy radiation emitted by radionuclides to eradicate tumor cells, thus achieving the therapeutic effect. Recently, with the continuous development of biomedical technology, the application of radionuclides in treatment of HCC has progressed steadily. This review focuses on three types of radionuclide-based treatment regimens, including transarterial radioembolization (TARE), radioactive seed implantation, and radioimmunotherapy. Their research progress and clinical applications are summarized. The advantages, limitations, and clinical potential of radionuclide treatment of HCC are discussed.

## 1. Introduction

Hepatocellular carcinoma (HCC) is the third leading cause of cancer death worldwide [1]. HCC is the main histologic type of primary liver cancer, accounting for 70–90% of liver cancer. Cirrhosis is the strongest risk factor of HCC, and the main causes of cirrhosis are chronic hepatitis B (HBV) or hepatitis C (HCV) virus infection, excessive alcohol consumption, and excessive dietary intake of aflatoxins [2,3]. Aflatoxin, a food contaminant produced by Aspergillus molds, has been shown to be an important pathogen in the pathogenesis of HCC. Increased aflatoxin intake is associated with the risk of HCC [4].

Clinical treatments for liver cancer mainly include surgical resection, liver transplantation, radiofrequency ablation (RFA), external radiation therapy, transcatheter arterial chemoembolization (TACE), and targeted drugs such as sorafenib. Sorafenib is a multiple-target tyrosine kinase inhibitor, which can inhibit RAF-1, B-Raf, and kinase activities in the Ras/Raf/MEK/ERK signaling pathway to inhibit tumor cell proliferation, and prolong the overall median survival of patients with advanced HCC [5]. According to the progression, the Barcelona Clinic Liver Cancer (BCLC) classification defines liver cancer into four stages: early (BCLC 0/A), middle (BCLC B), late (BCLC C), and terminal (BCLC D) [6]. For patients with early liver cancer or cirrhosis (BCLC grade 0 or A), surgical resection, liver transplantation, and radiofrequency ablation (RFA) are the main treatments. These treatments are effective and significantly prolong the survival of patients. However, liver cancer is usually asymptomatic or asymptomatic in the early stages. Therefore, most patients are in the middle or late stage at diagnosis, and they are not suitable for the above treatment protocols.

Unlike most cancers, HCC can be diagnosed by imaging without tissue sampling. MRI and CT are clinical methods used to diagnose HCC with an excellent performance. Dynamic MRI has slightly better diagnostic performance than CT imaging. CT has the advantages of a lower cost, higher availability, and faster scanning time [7]. For unresectable HCC (BCLC B), TACE may be used to deliver the drug to the tumor site via the hepatic artery. Considering that the liver cancer cells are mainly supplied by the hepatic artery, the treatment can effectively reduce the damage to normal liver tissue caused by the drug. Patients with TACE failure or BCLC grade C can be treated with systemic therapy agents such as sorafenib [8,9,10,11].

In addition to the above modalities, the therapeutic methods related to radionuclides represent an important research direction in the field of HCC treatment. The main radionuclide-related therapies for HCC include transarterial radioembolization (TARE), intratumoral implantation of radioactive particles, and radioimmunotherapy. The radionuclides commonly used in these treatments are ^131^iodine (^131^I), ^90^yttrium (^90^Y), ^188^rhenium (^188^Re), ^166^holmium (^166^Ho), and ^125^iodine(^125^I); the related studies and data are shown in Table 1 [12,13,14]. This review article introduces the above three therapeutic methods; summarizes the clinical application status and research progress of related radiopharmaceuticals; and discusses the advantages, limitations, and prospects of radionuclides in the treatment of HCC.

## 2. Transarterial Radioembolization

TARE is a new modality of radionuclide therapy of HCC [15]. Between 70% and 80% of the blood supply of liver tumors comes from the hepatic artery while normal liver tissue mainly relies on the portal vein for blood supply, with only 20%–30% coming from the hepatic artery [16,17]. According to the differences in the blood supply source between tumor tissue and normal liver tissue [18,19], the injection of radioactive agent into patients through the hepatic artery can deliver more radiation to the tumor site, thus reducing drug-induced hepatotoxicity [20,21].

### 2.1. TARE-Related Radiation Agents

The main radioactive agents used in TARE are ^131^I-lipiodol, ^90^Y-microspheres, ^188^re-lipiodol, and ^166^Ho-microspheres. The properties of these radionuclides are shown in Table 1. The beta rays emitted by these radionuclides break the double strands of DNA and kill surrounding cells. TARE allows the drug to be delivered to the tumor to kill more tumor cells and cause less damage to normal tissue [14,22,23,24]. Under normal circumstances, the radiation dose of external radiotherapy is 35 Gy, and the therapeutic effect is limited, with approximately 5% of patients going on to develop radiation-induced liver disease. Internal radiotherapy embolization can increase the radiation dose to 120 Gy or even higher, which not only effectively improves the therapeutic effect but also greatly reduces other side-effects caused by radiation [25,26]. ^131^I has a long physical half-life of 8.02 days and emits both beta and gamma rays. Patients require hospitalization after ^131^I injection for radiation protection after treatment. Although ^188^Re also emits beta and gamma rays, it has a 16.9 h half-life and emits fewer low-energy rays, which means that hospitalization after treatment is unnecessary. ^90^Y is a pure beta emitter with a physical half-life of 64.1 h. Patients can be discharged quickly after injection without radiological protection [27].

#### 2.1.1. ^131^I-Lipiodol

^131^I is the first radionuclide used for transcatheter arteria radioembolization. Lipiodol is a mixture of ethyl iodide of fatty acids from poppy seed oil, which typically contains 37% iodine. It is formed by replacing iodine in lipiodol with radioactive ^131^I through an atom–atom exchange reaction [28]. ^131^I-Lipiodol was first applied to humans in 1986 [29]. Intrahepatic arterial injection of ^131^I-lipiodol is selective and remains in tumors for a long time. Lipiodol is often used as a carrier of anticancer agent and contrast agents for radiography [30].

Studies have shown that ^131^I-lipiodol treatment is well tolerated. It has little toxic side effects and relieves patients’ pain to a certain extent. In recent studies, ^131^I-lipiodol has been used either as a single treatment or as an adjuvant treatment along with other regimens.

^131^I-Lipiodol as a treatment alone can effectively increase the survival rate of patients. In the study by Lintia-Gaultier et al., 50 patients with advanced liver cancer received ^131^I-lipiodol and 36 patients received only medical support. The 6-month, 1-year, and 2-year survival rates of patients in the ^131^I-lipiodol group were 65%, 35%, and 22%, respectively, while those in the control group were 28%, 8%, and 0%, respectively. The results indicate that ^131^I-lipiodol treatment significantly prolongs the survival time of patients with advanced HCC [28].

The combination of ^131^I-lipiodol therapy with other therapies also significantly prolongs the survival time of patients. In the study by Raoul et al., 34 patients were treated with ^131^I-lipiodol before liver surgery, among whom 25 showed an objective tumor response or histological necrosis of the major lesion site [31]. Boucher et al. conducted a retrospective study of patients treated with ^131^I-lipiodol after liver resection, and they found that treatment with ^131^I-lipiodol after surgery prolonged the disease-free and overall survival (Figure 1) [32].

Additionally, several studies have compared ^131^I-lipiodol therapy with non-radioactive lipiodol therapy. Lipiodol is a radiation carrier for the treatment of unresectable HCC, which is selectively injected into the hepatic artery of HCC patients. Lipiodol has prolonged retention in the tumor, but it shows no obvious anticancer effect. With the addition of ^131^I, ^131^I-lipiodol has been proven to be an effective therapeutic agent for HCC. The study by Dumortier et al. compared the efficacy of lipiodol and ^131^I-lipiodol. Patients with liver cancer (n = 58) were randomly treated with lipiodol or ^131^I-lipiodol within 6 weeks of tumor resection. The results showed that ^131^I-lipiodol effectively reduced the recurrence of HCC after hepatectomy, but no significant difference was found in improving the overall survival rate [33]. Moreover, Raoul et al. compared TACE and ^131^I-lipiodol. The results demonstrated that the 0.5-, 1-, 2-, 3-, and 4-year overall survival rates of patients treated with ^131^I-lipiodol were 69%, 38%, 22%, 14%, and 10%, respectively, while those of patients in the TACE group were 66%, 42%, 22%, 3%, and 0%, respectively. There was no significant difference in the patient survival between the two treatments, but patients treated with ^131^I-lipiodol showed better tolerance [29].

Most patients with HCC tolerate ^131^I-lipiodol therapy well, although interstitial pneumonia is a serious complication that may occur. According to the statistics of Jouneau et al., 15 of 1000 patients developed interstitial pneumonia after treatment and 12 of them died during 1994–2009 [34]. The above ^131^I-lipiodol-related studies and data are shown in Table 2.

Overall, when ^131^I-lipiodol is used as a radiopharmaceutical in the treatment of unresectable HCC patients for whom TACE or sorafenib is not appropriate, it can prolong disease-free survival, although its effect on overall survival is limited. Patients treated with ^131^I-lipiodol had a longer time from clinically confirmed complete remission to lesion recurrence, which greatly reduces the risk of tumor recurrence. Moreover, for patients waiting for liver transplantation, treatment with ^131^I-Lipiodol during the waiting period can slow tumor growth and metastasis and reduce the risk of being removed from the waiting list. Currently, the application of ^131^I-lipiodol for treatment HCC still needs the support of more effective clinical data.

#### 2.1.2. ^188^Re-Lipiodol

In the study of radionuclides applied in the medical field, ^188^Re is one of the ideal radionuclides used in treatment. It has a half-life of 16.9 h and emits both β and γ rays. Compared to ^131^I, ^188^Re has the advantages of a low price, no hospitalization and isolation after treatment, and it is more suitable for Asian and African countries [6]. At present, there are three types of ^188^Re-related preparations in clinical research, including ^188^Re-HDD lipiodol, ^188^Re-SSS lipiodol, and ^188^Re–DEDC lipiodol. Various methods of labeling lipiodol with ^188^Re have been proposed. So far, three different ^188^Re-labeled lipiodol complexes have been tested in humans, namely ^188^Re-HDD lipiodol, ^188^Re-SSS lipiodol, and ^188^Re-DEDC lipiodol. ^188^Re-HDD lipiodol is the most widely studied compound, but the in vivo stability of this complex is not optimal. Compared with ^188^Re-HDD lipiodol, ^188^Re-SSS lipiodol has superior in vivo stability. ^188^Re-DEDC lipiodol has been tested in animals and humans and showed prolonged retention in tumors with no significant release of the complex after in vivo administration [35].

The ^188^Re-HDD lipiodol Phase I and II clinical studies sponsored by the International Atomic Energy Agency (IAEA) evaluated the safety and efficacy of transarterial ^188^Re-HDD lipiodol for treatment-inoperable HCC. In the Phase I clinical trial, 70 patients received at least one ^188^Re-HDD lipiodol treatment and the results showed a median survival of 9.5 months [36]. The Phase II clinical trial results of the study, published in 2007, show that of the 185 patients from 8 countries who received ^188^Re iodine oil treatment, the 1-year and 2-year survival rates were 46% and 23%, respectively, with an observed good tolerance [37].

Kostas Delaunay et al. conducted a Phase I study of ^188^Re-SSS lipiodol for the treatment of HCC. The results show that ^188^Re-SSS lipiodol has a good biodistribution in radioactive embolism, and, of the radiolabeled lipiodols reported to date, it is the most stable in the body [38]. However, clinical studies of ^188^Re-DEDC lipiodol only show that it is safe and effective for treating inoperable HCC [39]. Further studies and clinical trial data are required to support the use of ^188^Re-related lipiodol in HCC. The above ^188^Re-lipiodol-related studies and data are shown in Table 3.

#### 2.1.3. ^90^Y-microspheres

^90^Y-microspheres was first used for tumor treatment in the 1960s [40], and it is the first radionuclide used for the treatment of HCC with portal vein thrombosis [41]. Clinical studies of ^90^Y-microspheres have been focused on bridging and downgrading in the middle and late stages of HCC and before liver transplantation [42,43]. Currently, ^90^Y-microspheres for the treatment of HCC are mainly made of glass and resin. ^90^Y-glass microspheres were approved by the Food Drug Administration (FDA) in 1999 for the adjuvant therapy of unresectable HCC and bridging liver transplantation, and it was later approved for the treatment of HCC with portal vein thrombosis. ^90^Y-resin microspheres were approved by the FDA in 2002 to be used along with fluorouridine for treating liver metastatic colorectal cancer [44,45]. ^90^Y-glass microspheres range from 20 to 30 microns, whereas ^90^Y-resin microspheres are usually 20 to 60 microns. The radiation activity of the ^90^Y-glass microspheres generally used is 2500 Bq while that of ^90^Y-resin microspheres is only 50 Bq [46,47].

In 2018, the American Association for the Study of Liver Diseases recommended the use of ^90^Y-microspheres TARE as the first-line treatment for HCC [15]. The institute determined the overall survival of patients with HCC who received ^90^Y-microspheres radionuclide embolization between 2003 and 2017 according to the BCLC staging. The overall survival of BCLC A, B, and C was 47.3 (39.5–80.3 months), 25.0 (17.3 to 30.5 months), and 15.0 months (13.8 to 17.7 months), respectively. The efficacy of the ^90^Y-microspheres TARE treatment for HCC has been confirmed by several studies. Hilgard et al. analyzed the data from 108 patients with advanced liver cancer and cirrhosis who received ^90^Y-microspheres TARE. According to the European Association for the Study of the Liver (EASL) criteria, the patients with a complete response, partial response, and disease stability accounted for 3%, 37%, and 53%, respectively, and 6% of the patients showed primary progression. The median progression time was 10 months and the median survival time was 6.4 months. In this study, the time to progression (TTP) and survival data of patients with advanced HCC were analyzed. It was found that the efficacy of ^90^Y-microspheres TARE was comparable to that of systemic therapy for patients with advanced HCC (Figure 2) [48]. D’Avola et al. demonstrated that ^90^Y-microspheres TARE extends the median survival compared to conventional care alone. This study compared the overall survival of 35 patients with unresectable HCC who received ^90^Y-microspheres treatment with 43 patients who received routine care only. The results showed that the median survival time was 16 months in the embolization group versus only 8 months in the control group [49].

Additionally, ^90^Y-microspheres TARE is used as an adjunctive therapy for preoperative bridging and degradation in patients awaiting liver transplantation. Gabr et al. performed a study of ^90^Y-microspheres for the treatment of liver transplantation patients from 2004 to 2018, among which 169 of 207 patients were treated with ^90^Y-microspheres TARE before liver transplantation, and another 38 patients received liver transplantation after staging was reduced by ^90^Y-microspheres TARE. According to the histopathology, 94 patients had complete necrosis of the tumor, accounting for 45% of the total patients; 60 patients had major necrosis of tumor tissue; and only 53 patients had local necrosis, accounting for 23%. The 3-, 5-, and 10-year survival rates were 84%, 77%, and 60% for patients with complete, major, and partial tumor necrosis, respectively. These results suggest that ^90^Y-microspheres TARE as an emerging adjunctive therapy is highly effective for bridging or reducing staging before liver transplantation [50]. Levi Sandri et al. also published similar data following a review of 20 global studies on ^90^Y-microspheres TARE as bridging and staging reduction for liver transplantation. A total of 178 patients were treated with ^90^Y-microspheres TARE before liver transplantation. The statistical results showed that ^90^Y-microspheres TARE was more effective than TACE in patients with advanced HCC (BCLC C) [51].

^90^Y-microspheres TARE is also used to treat patients with HCC with iatrogenic acute liver failure and portal vein thrombosis (PVT). ^90^Y-microspheres is the first radiopharmaceutical to be used for the treatment of HCC with PVT. According to the statistics of Ozkan et al., among 29 patients with HCC treated with ^90^Y-microspheres TARE between 2009 and 2014, PVT was formed in 12 patients, and the median survival was 17 ± 2.5 months. The statistical results showed that PVT formation is not an important factor affecting prognosis, and that ^90^Y-microspheres TARE treatment did not affect the median survival time of patients with PVT; however, TACE was contraindicated [52]. Similar results were found in a retrospective analysis published in 2010 by Inarrairaegui et al. The authors analyzed the data of 25 patients with PVT-formed HCC treated with ^90^Y-microspheres TARE. The statistical results demonstrated that the treatment of the PVT-formed HCC was well tolerated and had a favorable median survival. No hepatotoxicity was observed after 1–2 months of treatment, and the median survival of the patients was 10 months. However, the statistical results lacked further validation [53].

Whether ^90^Y-microspheres TARE combined with other methods is better than single therapy for HCC remains to be determined. Sorafenib and Micro-therapy Guided by Primovist Enhanced MRI in Patients With Inoperable Liver Cancer (SORAMIC) is a multicenter, randomized controlled trial for treating HCC that combines ^90^Y-microspheres TARE with sorafenib. In this study, a total of 424 patients with advanced HCC were randomized to ^90^Y-resin microspheres along with sorafenib treatment or sorafenib alone. The results showed that the median survival was 12.1 months in the combination group and 11.4 months in the other group, suggesting that the combination therapy showed no significant improvement regarding the survival of the patients [54].

Researchers have also tried combining this treatment with the PD-1 inhibitor in a clinical study. PD-1 inhibitors are important immunosuppressive molecules that help immune cells in the body recognize and kill tumors. Nivolumab, a PD-1 inhibitor approved by the FDA in 2015, is aimed at patients with advanced HCC who have been treated with sorafenib. In 2018, Wehrenberg-klee reported a case in which a patient was successfully bridged for partial hepatectomy using ^90^Y-microspheres TARE combined with PD-1 inhibitor therapy. The combined use of ^90^Y-microspheres with nivolumab or other immunotherapies may help improve the efficiency and degree of response to HCC therapy, enhance the ability to deliver radiation doses to tumors, and mediate other possible pro-inflammatory effects of embolism. Therefore, ^90^Y-microspheres TARE combined with immunotherapy may have an impact on advanced HCC [55].

Compared to TACE, ^90^Y-microspheres TARE does not significantly extend the total survival time of patients, but it is obviously superior to TACE in prolonging the time before progression. According to a Phase II clinical trial by Salem et al., between 2009 and 2015, 179 BCLC A or B patients with HCC were randomized to conventional TACE or ^90^Y-microspheres TARE. The results showed that the median progression time in the ^90^Y-microspheres TARE group was longer than 26 months while that in the TACE group was only 6.8 months [56]. Salem et al. retrospectively analyzed the data of 245 patients with HCC, including 122 who received TACE and 123 who received ^90^Y-microspheres TARE. The median progression time was 13.3 months in the ^90^Y-microsphere TARE group and 8.4 months in the TACE group while the median survival time was 20.5 months in the ^90^Y-microspheres TARE group and 17.4 months in the TACE group [57]. These studies showed that ^90^Y-microspheres TARE significantly prolongs the median progression time in patients with HCC.

Although ^90^Y-microspheres TARE has no significant improvement on the survival of patients compared to the traditional drug sorafenib, ^90^Y-microspheres TARE significantly increases the tumor response, reduces the occurrence of adverse events, and improves patients’ quality of life. This conclusion is supported by two large randomized controlled clinical trials. Chow et al. reported a Phase III trial in which 360 patients with HCC from 11 countries in the Asia-Pacific region were randomly assigned to be treated with ^90^Y-microspheres TARE or sorafenib. The results showed that the median survival was 8.8 months for patients treated with ^90^Y-microspheres TARE while that of patients treated with sorafenib was 10 months, indicating that there was no significant difference in extending the median survival in patients with locally advanced HCC [58]. Moreover, a Phase III clinical trial in Germany on advanced HCC with TARE examined 467 patients with advanced HCC who were randomized to receive ^90^Y resin-based microspheres or sorafenib treatment. The median survival time was 8 months for patients treated with ^90^Y resin-based microspheres TARE while that of patients treated with sorafenib was 9.9 months. The results demonstrate that there is no significant difference between the two treatments in extending the median survival of patients [59].

Patients with HCC who are treated with ^90^Y-microspheres TARE may have minimal adverse effects with less severe symptoms, including fatigue, nausea, vomiting, anorexia, fever, and abdominal discomfort; these symptoms are less likely to occur and generally do not require hospitalization. More serious symptoms include hepatic dysfunction, biliary toxicity, fibrosis, radiation pneumonitis, gastrointestinal complications, and vascular injury [44]. However, the probability of these serious side effects is extremely low, with less than 4% of liver disease cases being induced by radiation. According to Salem et al., less than 2% of patients require interventional therapy due to biliary toxicity induced by radioembolization, and the incidence of radiation pneumonitis induced by radioembolization is less than 1% [60,61,62,63]. Kallini et al. performed a retrospective analysis to determine whether there is a safety difference between ^90^Y-glass microspheres and ^90^Y-resin microspheres. A total of 1579 patients in 24 studies were treated with ^90^Y-glass microspheres, and 720 patients in 9 studies were treated with resin microspheres. The statistical results showed that compared to the ^90^Y-resin microspheres, ^90^Y-glass microspheres have a lower incidence of gastrointestinal and pulmonary adverse events for the treatment of HCC [64]. The ^90^Y-microspheres-TARE-related studies and data are detailed in Table 4.

^90^Y-microspheres TARE treatment is not significantly different from TACE or sorafenib treatment in terms of extending the overall survival in patients. TACE is used for bridging or degrading before liver transplantation, reducing the risk of patients being disqualified from transplantation due to tumor progression while waiting for liver transplantation. For patients with HCC with PVT, the replacement of TACE with ^90^Y-microspheres TARE does not affect the median survival. Patients with advanced HCC who are not responding to TACE or sorafenib may also be considered for treatment with ^90^Y-microspheres TARE. The phase of the use of ^90^Y-microspheres TARE in the standardized treatment of HCC is not clear yet, and there are also uncertainties about the prognostic effect of ^90^Y-microspheres TARE in different HCC patients.

#### 2.1.4. ^166^Ho-Microspheres

At present, there are three types of commercial radioactive microspheres, namely, ^90^Y-resin microspheres, ^90^Y-glass microspheres, and ^166^Ho-poly-l-lactic acid microspheres. ^166^Ho emits 81 keV gamma photons when it decays and is also a lanthanide element, and it can be imaged by single-photon emission computed tomography (SPECT)/magnetic resonance imaging (MRI) [65].

The Holmium Embolization Particles for Arterial Radiotherapy (HEPAR) trial is a Phase I clinical trial of ^166^Ho-microspheres, which eventually determined the maximum radiation dose tolerated by the ^166^Ho-microspheres to be 60 Gy [66]. Among the 37 patients in Phase II of the HEPAR trial, 73% of the patients showed complete remission, partial remission, or a stable condition after 3 months of treatment. Additionally, the adverse event rate is comparable to that of known ^90^Y-microspheres TARE therapy [67]. More Phase II trials of ^166^Ho-microspheres are underway.

## 3. Radioactive Seed Implantation

Radioactive seed implantation relies on stereoscopic imaging equipment to implant radioactive seed into the tumor for eradication by radiation. The research of ^125^I seed implantation for the treatment of HCC has increased in recent years.

The ^125^I seeds are prepared by wrapping a titanium alloy around a silver rod with ^125^I. This technique relies on B-scan ultrasonography, computed tomography (CT), MRI, and other imaging equipment to guide the ^125^I seed into the tumor tissue, through which the ^125^I seed continues to emit low-dose γ rays to treat the tumor. ^125^I has a long half-life of 60.1 days, which allows it to function continuously in tumor tissue. Additionally, the radiation distance of ^125^I is only 1.7 cm, which causes a low level of damage to normal tissue [68,69]. Recent studies of ^125^I seed implantation for the treatment of HCC have focused on the combination of other therapies. Among them, ^125^I seed implantation combined with TACE, RFA and surgical treatment, or treatment of PVT-formed HCC is the focus of research.

^125^I seed implantation combined with TACE therapy has received considerable attention, with some studies showing that ^125^I seed implantation combined with TACE is safe and effective for treating HCC, with a significantly prolonged total and progression-free survival time. Zhang et al. collected clinical data from 110 patients with advanced primary liver cancer from 2014 to 2016, among whom 55 patients received ^125^I seed implantation plus TACE and 3D conformal radiotherapy while the other 55 patients received TACE plus 3D conformal radiotherapy. The results showed that the objective remission rate of the ^125^I seed implantation plus TACE and 3D conformal radiotherapy group was 84% while the disease control rate was 96%. However, patients that only received TACE plus 3D conformal radiotherapy showed a conventional objective response rate of 64%, and the disease control rate was 84%, respectively. The results showed that ^125^I seed implantation combined with conventional treatment can significantly prolong the overall and progression-free survival [70]. In the work of Fang et al., 76 patients with HCC with PVT received TACE plus RFA or ^125^I seed implantation plus TACE and RFA treatment, respectively; the median survival was 30 and 42 months and the median progression-free period was 11 and 18 months, respectively. These results further validate the safety and efficacy of the combination of the three therapies [71].

Some patients with HCC can be treated with ^125^I seed implantation after RFA treatment; however, the effect of this combination therapy on patient survival rates has shown variability across studies. In a randomized trial by Chen et al., 136 patients with HCC were randomly divided into two groups: One group received ^125^I seed implantation therapy after RFA treatment while the other was treated with RFA only. The results showed that the survival rate of the RFA plus ^125^I seed implantation group was obviously better than that of the single RFA group [72]. However, a randomized controlled trial conducted by Wu et al. showed that the progression-free survival was 18 months in the combined treatment group, which was 7 months longer than that in the RFA group, but there was no significant difference in the overall survival between the two groups [73]. In another clinical trial of ^125^I seed implantation by Chen et al., 68 patients with HCC undergoing surgery were randomly assigned to receive ^125^I seed implantation or medical support. The relapse time of the two groups was 60 and 36.7 months, respectively, and the 1-, 3-, and 5-year survival rates were 94%, 74%, and 56%, and 88%, 53%, and 29%, respectively. The results showed that ^125^I seed implantation therapy after surgery can significantly prolong the disease-free survival and overall survival in patients with HCC [74]. Currently, the study of ^125^I seed implantation combined with RFA or surgical treatment is unsatisfactory, and more clinical data and statistical analysis are needed to obtain a clear conclusion on the effects on survival.

Additionally, studies have examined the use of ^125^I seed implantation to treat PVT-formed HCC. The available data only shows that it is safe for use in these patients but does not determine whether it is effective. According to the statistical results of research by Zhang et al. on six related studies, 406 patients with HCC with PVT received ^125^I seed implantation treatment. The side effects of radiation included leukopenia while the adverse reactions associated with ^125^I seed implantation included fever, abdominal pain, bleeding, and anorexia. No stent or particle migration was reported in these patients. The results indicated that the use of ^125^I seed implantation is safe in patients with HCC [75], but the efficacy of the treatment needs to be determined in more clinical trials. The relevant studies and data of ^125^I seed implantation are detailed in Table 5.

^125^I seed implantation has advantages, including less trauma, a uniform distribution in the tumor, less damage to normal tissue, reduced treatment time, fewer treatments, and no isolation after treatment. This approach can be used to treat inoperable HCC or PVT-formed HCC that does not respond to TACE or sorafenib treatment. However, based on the current studies, more clinical data are needed to support the safety and efficacy of ^125^I seed implantation.

## 4. Radioimmunotherapy

Radioimmunotherapy can be used as a means to treat tumors with radionuclide-labeled antibodies. HCC-targeted antibodies labeled with ^131^I have been intensively studied for the treatment of HCC, with the most common antibodies including mouse anti-human monoclonal antibody fragment HAb18F(ab)_2_ (metuximab), ChTNT human-mouse chimeric antibody, hepama-1 HCC cell membrane monoclonal antibody, CD133 monoclonal antibody, anti-hepatitis B virus antibodies, anti-machine protein monoclonal antibody, and anti-human HCC transferrin monoclonal antibody. Radioimmunotherapy agents used for HCC with clinical trials include ^131^I-metuximab, ^131^I-chTNT, and ^131^I-hepama-1 monoclonal antibody [76,77].

### 4.1. ^131^I-Metuximab

Metuximab is a mouse anti-human monoclonal antibody fragment HAb18F (ab)_2_, the antigen of which is HAb18G/CD147, which has high expression in liver cancer, colon cancer, and cervical cancer, among others. HAb18G/CD147 is a highly glycosylated cell surface transmembrane protein belonging to the immunoglobulin superfamily. It has been reported that the high expression of CD147 is closely related to the invasion, metastasis, and growth of tumors and is a significant independent predictor. It has been reported that blocking HAb18G/CD147 expression with ^131^I-metuximab effectively inhibits HCC growth and metastasis in vivo [78].

Studies on the safety and efficacy of ^131^I-metuximab for the treatment of HCC have shown no life-threatening toxicity. In a Phase I clinical trial published by et al., the safe dose of ^131^I-metuximab was 27.75 MBq/kg. In the subsequent multicenter Phase II trial, of 73 tracked patients, 6 showed partial remission (8%), 14 showed mild remission (19%), and 43 were in a stable condition (59%), with a 21-month survival rate of 45% [79].

Studies have shown that combined treatment with ^131^I-metuximab and TACE improved the survival and delayed recurrence in patients with unresectable HCC. Ma et al. conducted a Phase IV clinical trial of ^131^I-metuximab along with TACE for the treatment of inoperable HCC. In this multicenter, open-label clinical trial, 341 patients with stage III/IV HCC were non-randomly assigned to the trial group (n = 167) and the control group (n = 174) to receive combination therapy of ^131^I-metuximab plus TACE or TACE alone. It was found that ^131^I-metuximab combined with TACE improved the 1-year survival rate and prolonged the time of tumor progression, and the 1-year survival rate of the experimental group was 79.47% while that of the control group was 65.59%. The time of progression in the experimental group was 6.82 ± 1.28 months, which was approximately 2 months longer than that of the control group [80]. Similar results were found in the studies of He et al., in which 185 patients with unresectable HCC were treated with ^131^I-metuximab plus TACE (95) or with TACE alone (90). The 1-month effective rate was 71% in the trial group and 39% in the control group. The 6-, 9-, and 12-month survival rates in the combined treatment group were 86%, 74%, and 60%, respectively, while those in the control group were 60%, 42%, and 34%, respectively. The results of this study showed that the combination of ^131^I-metuximab plus TACE significantly increased the efficacy within 1 month and prolonged the survival of patients with HCC compared to those with TACE alone [81].

Delaying the recurrence of HCC is the key to the treatment of HCC. Treatment with ^131^I- metuximab after liver transplantation or RFA is helpful to reduce recurrence. In the study by Xu et al., 60 patients with HCC with liver transplantation were randomly divided into two groups. The treatment group received ^131^I-metuximab at 15.4 MBq/kg 3 weeks after liver transplantation and the control group was given a placebo intravenously. At the 1-year follow-up, compared to the control group, the recurrence rate was significantly reduced by 30% and the survival rate increased by 21% in the treatment group. The results showed that ^131^I-metuximab is effective in reducing tumor recurrence and improving the survival rate in patients with HCC after transplantation [82]. Moreover, Bian et al. evaluated the efficacy of ^131^I-metuximab along with RFA for the treatment of HCC. In this study, 127 patients with HCC with stage 0-B BCLC were randomly divided into two groups. One group received RFA followed up with ^131^I-metuximab while the other group received only RFA. The results showed that the 1- and 2-year recurrence rates were 32% and 59% in the combined group and 56% and 71% in the RFA group, respectively. The median time of recurrence was 17 and 10 months in both groups. The results of this study suggest that the use of ^131^I-metuximab after RFA may be helpful in the prevention of postoperative recurrence [83].

### 4.2. ^131^I-chTNT

ChTNT is a mouse chimeric antibody. When labeled with ^131^I, the ^131^I-chTNT antibody binds to intracellular antigens in the necrotic part of the tumor. Intracellular antigen is a complex of double-stranded DNA and histone H1 antigen that is present in scattered areas of degenerated or necrotic cells within a tumor. The antibodies commonly used in targeted therapies primarily bind to antigens on the surface of tumor cells, but TNT antibodies can bind to intracellular antigens at the site of tumor necrosis. ^131^I acts to treat the surrounding tumor cells, causing new necrosis, while the chTNT monoclonal antibody expands to the newly necrotic area to continuously expand it to achieve the therapeutic goal. At present, ^131^I-chTNT is considered to have a therapeutic effect on lung cancer, brain cancer, and liver cancer, among others [76,80].

Data from patients with HCC treated with ^131^I-chTNT were retrospectively analyzed by Tu et al. Among 38 patients with HCC, 22 were treated with RFA only while the other 16 patients were treated with RFA plus ^131^I-chTNT. The median survival of the two groups was 37 and 43 months, respectively, while the 1-, 2-, and 3-year overall survival rates were 100%, 88%, and 75% (RFA plus ^131^I-chTNT), and 82%, 58%, and 52% (RFA). The retrospective analysis showed that RFA combined with ^131^I-CHTNT prolongs disease-free survival in the short term, better than RFA alone. However, a randomized controlled trial with a larger sample is needed to assess the efficacy of the treatment [84].

### 4.3. ^131^I-Hepama-1 mAb

Hepama-1 is a monoclonal antibody against the HCC cell membrane. HAb18G/CD147 is a highly glycosylated cell surface transmembrane protein belonging to the immunoglobulin superfamily. It has been reported that the high expression of CD147 is closely related to the invasion, metastasis, and growth of tumors and is a significant independent predictor. It has been reported that blocking HAb18G/CD147 expression with ^131^I-metuximab effectively inhibits HCC growth and metastasis in vivo [85]. Several studies in the late 1990s investigated the value of ^131^I-hepama-1 monoclonal antibodies in treating HCC. A Phase I trial conducted by Chen et al. treated 45 patients with HCC who could not be treated surgically with ^131^I-hepama-1 mAb. The results demonstrate that ^131^I-hepama-1 mAb is safe by intravenous injection, and the recommended dose of ^131^I-hepama-1 mAb is 1480–2960 MBq/10 mg [86]. The accompanying radioimmunoassay-related studies and data are detailed in Table 6.

Through an extensive literature review, it was found that the efficacy of radioimmunotherapy for the treatment of solid tumors needs to be improved. On this occasion, some in vivo studies have demonstrated the safety of radioimmunotherapy. For patients with HCC who are unamenable to surgical resection or monotherapy, a combination of radioimmunotherapy may be considered. The efficacy of the treatment is affected by several factors, including the targeting ability of monoclonal antibodies, the stability of the radioimmunoconjugate in vivo, and the mode of administration. The development of more targeted monoclonal antibodies for HCC, improvement of the radiochemical stability of the radiolabeled MAbs, and identification of suitable administration routes are future directions that require further investigations.

## 5. Summary and Future Prospects

TARE is well-tolerated and has few side effects in the treatment of advanced HCC. Although it has no obvious survival benefit compared to TACE or sorafenib in clinical trials, TARE can prolong disease-free survival and improve patients’ quality of life. Moreover, TARE may be considered in cases with PVT formation, failure of TACE/sorafenib therapy, bridging liver transplantation, or reduced-grade liver transplantation. ^90^Y-microspheres TARE is one of the most promising approaches of translating radionuclide therapy for HCC into routine treatment practice. However, the proper use of ^90^Y-microspheres TARE in the standardized treatment of HCC has not been cleared, and there are also uncertainties about the prognostic effect of ^90^Y-microspheres TARE in different HCC patients. ^125^I seed implantation and ^131^I-metuximab radioimmunotherapy for HCC have gained increasing attention, but their efficacy requires clinical validation by further randomized controlled trials. Clinical trials on nuclide treatment for liver cancer have mainly included TARE and radioactive seed implantation until now. There are 21 ongoing clinical trials of TARE for liver cancer, mainly on ^90^Y TARE. ^90^Y TARE has been demonstrated to be more effective and less toxic than TACE. In addition, clinical trials on the ^166^Ho radio-embolism and the combination of sorafenib with ^90^Y radio-embolism are also underway. There are five clinical trials on radioactive seed implantation, mainly conducted around ^125^I. The above information about clinical trials comes from Clinicaltrials.gov. The use of radionuclides carries a certain risk to medical staff, and how to regulate the operation during treatment for risk mitigation is worthy of attention.

## Figures and Tables

**Figure 1 pharmaceuticals-15-01339-f001:**
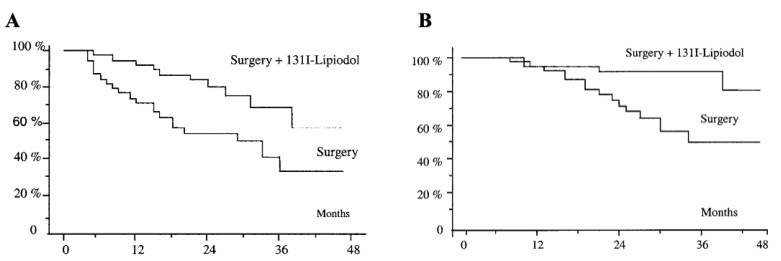
Adjuvant intra-arterial injection of iodine-131-labeled lipiodol after resection of HCC. (**A**) Disease-free survival of patients in the 2 treatment groups (*p* < 0.02). (**B**) Overall survival of patients in the 2 treatment groups (*p* < 0.02). Adapted with permission from [32].

**Figure 2 pharmaceuticals-15-01339-f002:**
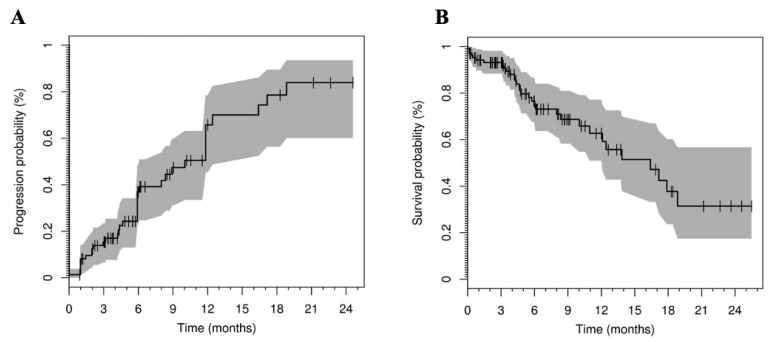
Radioembolization with yttrium-90 glass microspheres in hepatocellular carcinoma. (**A**) TTP (with progression defined according to RECIST with the recent NCI amendments) in 76 of 108 HCC patients treated by Y-90 glass microsphere radioembolization for which radiological response data were available. The solid line displays the Kaplan–Meier estimator, with marks representing censored events. The shaded area marks the limits of the pointwise 95% CIs. (**B**) Overall survival in 108 HCC patients treated by Y-90 glass microsphere radioembolization. The solid line displays the Kaplan–Meier estimator, with marks representing censored events. The shaded area marks the limits of the pointwise 95% CIs. Adapted with permission from [48].

**Table 1 pharmaceuticals-15-01339-t001:** Properties of radionuclides commonly used for TARE.

Radionuclides	Ray Species	Half-Life (Days)	Mean Tissue Penetration Depth (mm)
^131^I	γ, β-	8.04	0.4
^90^Y	β-	2.7	3
^188^Re	γ, β-	0.708	4.8
^166^Ho	γ, β-	1.116	2.5
^125^I	X-ray, γ	60.1	20

**Table 2 pharmaceuticals-15-01339-t002:** Advances for treatment of HCC using ^131^I-lipiodol TARE.

Author	Year	Experiment Type	Design	Number of Patients	Conclusion	Refs.
Jean-Luc Raoul	1997	Random control	^131^I-Lipiodol vs. TACE	129	Better tolerance, but no significant difference in OS	[29]
J-L Raoul	2003	Retrospective analysis	Use ^131^I-lipiodol before surgery	34	Tolerability and tumor response were good	[31]
Eveline Boucher	2003	Retrospective analysis	^131^I-Lipiodol after surgery vs. surgery	38	DFS and OS were better than the control group	[32]
Stéphane J Juneau	2011	Retrospective analysis	^131^I-Lipiodol treating induced interstitial lung disease for HCC	1000	Of 15 people with interstitial pneumonia, 12 died	[34]
Alina Lintia-Gaultier	2013	Retrospective comparative analysis	^131^I-Lipiodol vs. medical support	86	Overall survival (OS) was better than the control group	[28]
Jérôme Dumortier	2014	Random control	^131^I-Lipiodol vs. lipiodol	58	Reduced postoperative recurrence, but no significant difference in OS	[33]

**Table 3 pharmaceuticals-15-01339-t003:** Advances for the treatment of HCC with ^188^Re-lipiodol TARE.

Author	Year	Experiment Type	Design	Number of Patients	Conclusion	Refs.
Felix Sundram	2004	Clinical Phase I	^188^Re-HDD lipiodol treatment	70	Confirm dosage 98 GBq (265 mCi); Safe and minimal side effects	[35]
Patricia Bernal, MD	2007	Clinical Phase II	^188^Re-HDD lipiodol treatment	185	Well-tolerated; 1-year survival rate of 46%, 2-year survival rate of 23%	[36]
Kostas Delaunay	2019	Clinical Phase I	^188^Re-SSS lipiodol treatment	6	Good biodistribution and high stability in vivo	[37]

**Table 4 pharmaceuticals-15-01339-t004:** Advances for the treatment of HCC with ^90^Y-microspheres TARE.

Author	Year	Experiment Type	Design	Number of Patients	Conclusion	Refs.
Delia D’Avola	2009	Retrospective comparative analysis	^90^Y-microspheres vs. usual care	78	Median survival was significantly higher than the usual care	[49]
Philip Hilgard	2010	Random control	^90^Y-glass microspheres for advanced HCC and liver cirrhosis	108	Comparable efficacy to systemic therapy	[48]
Mercedes Inarrairaegui	2010	Retrospective analysis	^90^Y-microspheres TARE for HCC with PVT	25	Well-tolerated, favorable median survival	[53]
Riad Salem	2011	Retrospective analysis	^90^Y-microspheres TARE vs. TACE	245	The ^90^Y group showed an extended progression time, with no significant difference in the median survival	[57]
Zeynep Gozde Ozkan	2015	Retrospective analysis	^90^Y-microspheres TARE for HCC	29	The median survival time of patients with HCC was unaffected	[52]
Giovanni Battista Levi Sandri	2017	Retrospective analysis	^90^Y-microspheres TARE before transplantation	178	^90^Y TARE was better than TACE	[51]
Riad Salem	2016	Random control	^90^Y-microspheres TARE vs. TACE	179	^90^Y group had a longer median progress time	[56]
Valérie Vilgrain	2017	Clinical Phase III	^90^Y-microspheres TARE vs. sorafenib	467	No significant difference in OS	[59]
Joseph Ralph Kallini	2017	Retrospective comparative analysis	^90^Y-resin microspheres or ^90^Y-glass microspheres for HCC	2299	Glass microspheres in HCC treatment of gastrointestinal tract and lungs a lower incidence of adverse events	[64]
Eric Wehrenberg-Klee	2018	Case report	^90^Y-microspheres TARE + PD-1	1	Successful bridge partial liver resection surgery	[55]
Pkh Chow	2018	Clinical Phase III	^90^Y-microspheres TARE vs. sorafenib	360	No significant difference in OS	[58]
Jens Ricke	2019	Random control	^90^Y-resin microspheres + sorafenib vs. sorafenib	424	Combination therapy did not significantly improve the patients’ survival	[54]
Ahmed Gabr	2021	prospective study	Treatment with ^90^Y-microspheres adjuvant before liver transplantation	207	Support ^90^Y as a neoadjuvant therapy for bridging or decreasing staging before liver transplantation	[50]

**Table 5 pharmaceuticals-15-01339-t005:** Advances for the treatment of HCC using ^125^I seed implantation.

Author	Year	Experiment Type	Design	Number of Patients	Conclusion	Refs.
Kaiyun Chen	2013	Random control	After HCC, ^125^I seed implantation vs. medical support	68	Disease-free survival and OS were prolonged in the experimental group	[74]
Kaiyun Chen	2014	Random control	RFA +^125^I seed implantation vs. RFA	136	The survival rate of the RFA + ^125^I group was significantly better than that of the RFA group	[72]
F Z Wu	2016	Random control	RFA + ^125^I seed implantation vs. RFA	47	No significant difference in OS	[73]
S J Fang	2019	Retrospective comparative analysis	^125^I seed implantation+ TACE + RFA vs. TACE + RFA	76	Safety and efficacy of three therapeutic modalities in combination	[68]
Lei Zhang	2020	Random control	Patient with HCC with PVT accepted the ^125^I seed implantation	406	^125^I implantation is safe	[75]
Huanyun Zhang	2020	Random control	TACE + ^125^I seed implantation + 3D conformal radiotherapy vs. TACE + 3D conformal radiotherapy	110	Overall and progression-free survival were significantly prolonged	[70]

**Table 6 pharmaceuticals-15-01339-t006:** Advances in radioimmunotherapy for the treatment of HCC.

Author	Year	Experiment Type	Design	Number of Patients	Conclusion	Refs.
Chen S	2004	Clinical Phase I	For patients with unresectable HCC give ^131^I-hepama-1 MAb	45	The recommended dose for clinical use is 1480–2960 MBq/10 mg	[86]
Zhi-Nan Chen	2006	Clinical Phase I	^131^I-metuximab	28, 106	Safe dosage 27.75 MBq/kg; the survival rate of 21 months was 4454	[79]
Jing Xu	2007	Random control	Post-transplant,^131^I-metuximab vs. placebo	60	The recurrence rate decreased while the survival rate increased	[82]
He Q	2013	Nonrandomized prospective cohort study	^131^I-metuximab + TACE vs. TACE	185	The survival time of the patients in the combined treatment group was prolonged	[81]
Bian H	2014	Random control	Follow-up with ^131^I-metuximab after RFA treatment vs. RFA	127	Follow-up of ^131^I cetuximab after RFA treatment helped prevent postoperative recurrence	[83]
Tu J	2014	Retrospective comparative analysis	RFA+^131^I-chTNT vs. RFA	38	Combination therapy extended the disease-free survival of patients	[74]
Ma J	2015	Clinical Phase IV	^131^I-Metoximab + TACE vs. TACE	341	The 1-year survival and progression time of the experimental group were longer than those of the control group	[80]

## Data Availability

Data sharing not applicable.

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
