# Peer review of "Recent Advances in Hepatocellular Carcinoma Treatment with Radionuclides"

_pharmaceuticals, 2022, doi:10.3390/ph15111339_

Round 1

Reviewer 1 Report

The authors have prepared an interesting review of recent advances in the treatment of HCC with radionuclides. The authors have covered the literature well and the only complaint I have is that in some cases a little extra context could be given or some mention on certain attributes of radionuclide therapy could be provided. Otherwise, I recommend this manuscript for publication after minor revisions.

Please check all table to make sure entries are complete: e.g. Experiment type is not provided for all table entries in Table 2 and related tables.

Introduction

Line 33: Define aflatoxins as not all readers may know that those are derived from mold

Line 35: When mentioning Sorafenib for the first time give its target/mechanism of action; RAF kinase inhibitor

Line 46: Specify whether it is beta - or beta + as there are two type of beta decay

Line 47: Give the name of the element prior to the use of the elemental symbol  

Line 61: It does not just kill tumor cells, it predominately kills tumor cells but is not specific, please make your statement more precise

Table 1: State decay mode as Beta- and not just Beta

Line 93: In the comparison of Lipiodol with 131I-Lipiodol the difference between the radioisotope and naturally occurring iodine Lipiodol is not provided. This was an interesting control reaction, knowing the difference would be good to provide it is available in the reference.

Line 109: "... limited effect in extending overall survival, it extends the disease-free survival of patients" Could you explain the difference for these two metrics for the non-clinical scientists that will read this review.

1.1.2 188Re Three different versions of this were utilized, could you explain the difference between HDD, SSS, DEDC, provide figure with structure of chelator for Re if appropriate.

Line 191: "has no significantly improvement" please correct this typo

Line 192: Which PD-1 inhibitor was utilized, why was that particular one selected (you later mention Nivolumab, please make it clear from the start what the agent utilized was in the study)?

Line 207: "no significantly improvement" please correct this typo

Line 240: "...and the specific prognostic factors following treatment with 90Y-microsphere TARE" This should likely be its own sentence/thought as the meaning is unclear from how the start of the sentence is phrased. Please make this more clear and easy to understand.

Line 257: with the use iodine-125 it is possible Auger electrons could be responsible for some of the therapeutic effect, please provide some commentary on if Auger electrons from 125I are a factor in the results.

Line 301: Check heading formatting and tabs throughout  

Radioimmunotherapy Section: Provide some more info on the target selection for antibody, target type and expression in HCC versus healthy tissue. In addition, does HCC express the targets found with approved new radiotherapy agents like Pluvicto (PSMA) or Lutathera (SSTR2)? It would be interesting to know if those targets are expressed or not in HCC.

Line 307: Why just Iodine-131 and not other therapeutic radionuclides like lutetium-177, yittrium-90 or alpha emitters like Radium-223 or Actinium-225, provide some commentary if these are under consideration or possibilities in the future.

Line 310: Provide more info on CD147 as a target

Line 314: what is the %ID/g or some other measure of uptake in the tumor versus major organ (largest uptake other than the tumor). Is kidney toxicity observed and was the administration of amino acids considered or utilized to protect the kidneys as part of the study as is done with other radiotherapeutics?

Line 343: more info on "intracellular antigens" which ones and why.

With the above corrections I think the review will be improved and will be a means for the reader to understand where things currently stand in the treatment of HCC with radionuclides and in context with other ongoing radiotherapy efforts in oncology.

Author Response

Please see the attachment。

Reviewer 2 Report

General comments

In overall, the manuscript provides a comprehensive review of clinical studies aimed to establish the efficiency of radionuclide therapy of liver cancer. The review covers a wide range of radionuclides and delivery technologies exploited in this type of radiotherapy. The paper will be of great assistance for radiation oncologist to navigate through published studies in the area, as well as for radiation biologist developing new radionuclide treatment technologies. However multiple textual mistakes make the manuscript untidy and difficult to read. Addressing a few specific issues, as well as textual errors outlined below will improve the manuscript.

Specific comments

1.       Table 1 is not comprehensive. Given that some radionuclides emit more than one type of radiation, it would be informative to specify which type, or both, is exploited in radionuclide therapy, and for which type the mean penetration depth is given. Indicating the energy of radiation would also be useful. As for 125-I, it is also an Auger electron emitter. Is Auger electron emission important or irrelevant for radionuclide therapy?

2.       Page 3 line 69: what is “an isotope transformation reaction”. The referenced article [24] doesn’t provide a description for radioiodination.

3.       Page 4 Table 2: the authors state in lines 99-101, that “…interstitial pneumonia is a serious complication that may occur” after 131-I-lipiodol treatment, however in Table 2 they refer to usage of “131I-Lipiodol for treating induced interstitial lung disease”. Please clarify this discrepancy, namely what disease was treated: HCC of lung disease?

4.       Table 3: please follow the convention for units of measurement, GBq but not GBQs (plural is never used with units), mCi but not MCI (prefix M means 10^6, prefix m means 10^-3). Same in line 358 and Table 6, please check carefully units, 2775 mBq/kg is nothing in terms of dosage. More generally, please specify dosage/injected activity for all discussed results where possible. This parameter is very important.

5.       Lines 349-350: Do you mean that combination RFA plus 131I-chTNT reduces 1-, 2- and 3-year overall survival compared to RFA only?

Minor comments

Below are just a few required corrections, please read and check the text thoroughly, and fix numerous errors, since it is not a job for a reviewer to find and fix all your textual mistakes

1.       Please check and insert a space before citations in Introduction.

2.       Page 1 line 36: please fix the sentence, is it “…classification is defined into four stages…” or classification defines four stages of cancer?

3.       Page 3 line 283:  the number of significant digits (say 94.12%, four digits) is too excessive, such a precision is not realistic. Same for page 13 line 283 and others. Suggesting to round the numbers, regardless of how they are given in original articles.

4.       Page 9 line 207: please fix “…TARE has no significantly improvement...”, should it be “significant”? Same on page 11 line 234: “…TARE treatment has no significant different...” is grammatical nonsense. Also, the sentence must start with capital “Microspheres”.

5.       Lines 304 and 310: please be consistent with HAb18F and HAB18F.

6.       Line 330: please use acronym RFA (radiofrequency ablation) since you have introduced it.

7.       Line 343: the sentence “ChTNT is a mouse chimeric antibody, when labeled with 131I, the TNT antibody binds to intracellular antigens in the necrotic 343 part of the tumor.” Is the nonsense. Do you mean that TNT antibody binds to its antigen only when labelled with 131-I?

Reviewer 3 Report

The manuscript is well-written. I have only few comments to provide.

1) In the introduction, please provide some notions about diagnosis of hepatocellular carcinoma.

2)Please provide some information about elimination of radiopharmaceuticals, especially for  131I-Lipiodol.

3) as regards future perspectives, please provide some information regarding clinical trials in progress.
